# Structural Determinant of β-Amyloid Formation: From Transmembrane Protein Dimerization to β-Amyloid Aggregates

**DOI:** 10.3390/biomedicines10112753

**Published:** 2022-10-29

**Authors:** Nicolas Papadopoulos, Nuria Suelves, Florian Perrin, Devkee M. Vadukul, Céline Vrancx, Stefan N. Constantinescu, Pascal Kienlen-Campard

**Affiliations:** 1SIGN Unit, de Duve Institute, UCLouvain, 1200 Brussels, Belgium; 2Ludwig Institute for Cancer Research Brussels, 1348 Brussels, Belgium; 3Aging and Dementia Research Group, Cellular and Molecular (CEMO) Division, Institute of Neuroscience, UCLouvain, 1200 Brussels, Belgium; 4Memory Disorders Unit, Department of Neurology, Massachusetts General Hospital, Harvard Medical School, Boston, MA 02115, USA; 5Department of Chemistry, Molecular Sciences Research Hub, Imperial College London, London SW7 2BX, UK; 6Laboratory for Membrane Trafficking, VIB-Center for Brain and Disease Research, KU Leuven, 3000 Leuven, Belgium; 7Department of Neurosciences, KU Leuven, 3000 Leuven, Belgium; 8Walloon Excellence in Life Sciences and Biotechnology (WELBIO), 1300 Wavre, Belgium; 9Nuffield Department of Medicine, Ludwig Institute for Cancer Research, Oxford University, Oxford OX1 2JD, UK

**Keywords:** Alzheimer’s disease, Amyloid Precursor Protein, amyloid beta, *APP*-C99, dimerization, orientations, aggregation, oligomerization

## Abstract

Most neurodegenerative diseases have the characteristics of protein folding disorders, i.e., they cause lesions to appear in vulnerable regions of the nervous system, corresponding to protein aggregates that progressively spread through the neuronal network as the symptoms progress. Alzheimer’s disease is one of these diseases. It is characterized by two types of lesions: neurofibrillary tangles (NFTs) composed of tau proteins and senile plaques, formed essentially of amyloid peptides (Aβ). A combination of factors ranging from genetic mutations to age-related changes in the cellular context converge in this disease to accelerate Aβ deposition. Over the last two decades, numerous studies have attempted to elucidate how structural determinants of its precursor (*APP*) modify Aβ production, and to understand the processes leading to the formation of different Aβ aggregates, e.g., fibrils and oligomers. The synthesis proposed in this review indicates that the same motifs can control *APP* function and Aβ production essentially by regulating membrane protein dimerization, and subsequently Aβ aggregation processes. The distinct properties of these motifs and the cellular context regulate the *APP* conformation to trigger the transition to the amyloid pathology. This concept is critical to better decipher the patterns switching *APP* protein conformation from physiological to pathological and improve our understanding of the mechanisms underpinning the formation of amyloid fibrils that devastate neuronal functions.

## 1. Introduction

Many neurodegenerative diseases are classified into the family of protein folding (or protein conformation) diseases, which are characterized by the abnormal accumulation of aberrantly folded or unfolded proteins [1,2,3]. Alzheimer’s disease (AD) is a typical example of protein folding disease where protein misfolding and aggregation are considered to play a causative role [1,2,3].

The concept of protein folding disorder has its origins in the mid-19th century, when, in 1854, Rudolf Virchow coined the term amyloid, from the Latin word “amylum” (starch), to describe a substance in the starchy brain bodies that exhibited a chemical reaction resembling that of cellulose [4]. Indeed, AD is associated with the formation of highly organized fibrillar aggregates called amyloid or senile plaques [5]. These aggregates are β-sheet-rich structures and are formed by the aggregation of amyloid beta (Aβ) peptides that are produced by the processing of the Amyloid Precursor Protein (*APP*) [6]. Remarkably, brain-derived Aβ fibrils are highly polymorphic [6,7] but share a common right-hand twist, unlike the left-hand twist usually observed in fibrils produced in vitro [6,7,8]. Each fibril usually consists of multiple twisted protofilaments that show a so-called cross-β structure, in which the strands of a β-sheet run perpendicular to the long axis of the fibril. Additionally, the formation of β-amyloid fibrils in AD is preceded by the generation of soluble Aβ oligomeric intermediates that have been shown to mediate cellular toxicity in AD [9].

The formation of Aβ fibrils and oligomers can be promoted through pathological conformational changes, the underlying mechanisms of which are not yet fully understood. Several risk factors have been demonstrated to increase the propensity of these vulnerable proteins to self-associate. For example, gene mutations have been linked to the onset of inherited forms of AD [10] and it has been proven that they can affect the conformational dynamics of Aβ. Changes in environmental conditions, such as temperature, pressure, pH, and concentration of organic solvents, can also affect the aggregation propensity of amyloid-forming proteins [5]. An imbalance in protein production or clearing mechanisms can also lead to the accumulation of the amyloidogenic protein/peptide, indicating that misfolding processes are protein-concentration dependent [11]. Although certain factors such as mutations are present in the individual at the outset, the disease only develops over time. Ageing is therefore a key factor that contributes to the context leading to protein misfolding and aggregation. This is illustrated by the progressive build-up of cellular stressors (such as reactive oxygen species or toxins) that might perturb the cell proteostasis and therefore increase the accumulation of misfolded or unfolded proteins [12].

Among neurodegenerative disorders, AD is the most prevalent and the most frequent form of amyloidosis. AD patients show a progressive loss of memory and other cognitive functions, and AD accounts for over 60 percent of dementia diagnoses. The disease is also characterized by the gradual appearance of brain dysfunction and neurodegeneration, and defining pathological features consisting in amyloid deposits and neurofibrillary tangles (NFTs), which are intraneuronal aggregates primarily composed of hyperphosphorylated tau proteins. The gradual progression of protein aggregates allowed a staging to be set that classifies the degree of pathology by correlating the density of lesions in specific brain regions and clinical symptoms [13,14]. From a molecular point of view, our knowledge of the mechanisms underpinning AD pathogeny is mostly rooted in the amyloid cascade hypothesis (ACH), which states that an early alteration in Aβ production, clearance, or deposition is central in the etiology of AD [15,16]. The debate around the ACH has only heated up in recent years [17], mainly because of the failure of compounds targeting amyloid pathology to restore cognitive function when tested in clinical trials. Still, the correlation between Aβ deposition and the progression of AD is supported by undeniable arguments combining biochemical, anatomopathological, and genetic evidence. AD causal mutations (responsible for autosomal-dominant AD or ADAD) are located in the genes encoding the Amyloid Precursor Protein (*APP*) or Presenilin 1 and 2 (*PSEN1* and *PSEN2*, respectively) [18]. They trigger the accumulation of the Aβ peptide, which has a high propensity to aggregate, eventually forming the extracellular senile plaques. These insoluble assemblies gradually accumulate, explaining the characteristic progressive nature of the neurodegenerative process and the commonly late onset of clinical symptoms. In this review, we set to provide a structural perspective on the determinants that drive the generation of pathogenic Aβ peptides.

## 2. The Amyloid Precursor Protein Family and Aβ Production

The Amyloid Precursor Protein (*APP*) belongs to the *APP* protein family which includes *APP*, *APLP1*, and *APLP2* in mammals [19,20,21,22], *APPa* and *APPb* in zebrafish [23], and *APPL* in drosophila [24]. The evolution of the *APP* gene family (for a review see [25]) indicates that the functions of *APP*, *APLP1*, and *APLP2* have diverged to contribute distinctly to neuronal processes. After the discovery that Aβ was the main component of AD amyloid fibrils [26] and subsequent identification of *APP* [19,27] and *APLPs* [22,28,29] genes, extensive studies in knock-out models brought the assumption that *APP*, *APLP1*, and *APLP2* are partially functionally redundant (for a review see [30]). The assumption that *APP* and *APLPs* have overlapping functions is also supported by the similarities in their expression pattern, biochemical processing, and structure properties. All members of the *APP* family possess a large ectodomain composed of the E1 and E2 domains. Besides the E2 domain [31,32,33], the structures of the *APP* protein family have only partially been solved by conventional techniques. Yet, the recent advances in structure prediction algorithm led by AlphaFold [34] allow a reasonable approximation of the respective structures of *APP*, *APLP1*, and *APLP2*. Figure 1 depicts the predicted structures of each segment of the *APP* and *APLP* proteins. Close analysis reveals that isolated fragments of the extracellular domains are almost identical in structure, with the inter-fragments predicted to be unstructured. The E1 domain is composed of two separate segments, corresponding to the Growth Factor-Like Domain (GFLD) and copper-binding domain (CuBD). GFLD is made of two β-sheets separated by an α-helix and the CuBD consists of a three-stranded antiparallel β-sheet [35]. The E2 domain is formed by two coiled-coil α-helices which are conserved in all three members of the *APP* family [32,33] (Figure 1). The Kunitz-type protease inhibitor (KPI) domain is specific to *APLP2* and long isoforms of *APP* (*APP*_770_ and *APP*_751_) and not present in the neuronal isoform (*APP*_695_). It is composed of double-stranded β-sheets with a small α-helix [36]. Functionally, the extracellular domain of *APP/APLPs* binds components of the extracellular matrix such as heparin and collagen via the E1 and E2 domain [37,38]. Interaction with heparin plays a role in neurite outgrowth [39], *APP/APLPs* dimerization [32,38,40], and cell-cell adhesion [41]. The E1 and E2 domains are also reported to bind zinc and copper [42,43] via the E1 copper/zinc binding domain (Cu/Zn BD) (Figure 1) and less well-defined sites of the E2 domain. Binding of zinc and copper regulates *APP* and *APLPs cis* and *trans*-dimerization, possibly by regulating their interaction with heparin [44,45,46,47]. Interestingly, the major neuronal isoform of *APP* (*APP*_695_) [48] does not contain the KPI domain, whereas longer isoforms that contain the KPI are expressed in the periphery and, to a significant level, in platelets, in agreement with the reported role of *APP* KPI as an inhibitor of multiple coagulation factors [49].

The transmembrane (TM) domains of *APP* and *APLPs* are formed of a single α-helix and the intracellular domain is mostly unfolded with the exception of the C-terminus which is composed of two small α-helices (Figure 1). The similarity between *APP/APLPs* and Notch processing (see below) led some authors to hypothesize that the *APP* and *APLPs* [50] Intracellular C-terminal Domain (AICD) could possess signaling functions (see [51,52] for a review). Reporter gene assays based on Gal4-fusion constructs suggested that AICD could act as transcriptional regulator [53,54], but the controversy about the transcriptional activity of AICD and its target genes has only grown since then [55,56].

### Sequential Processing of the APP and APLPs and Production of β-Amyloid Peptides

The structural similarities between *APP* family members goes hand in hand with the overlapping functions of *APP*, *APLP1*, and *APLP2*. Yet, *APP* is the only member of the family giving rise to the formation of pathogenic Aβ aggregates. Members of the *APP* family are processed via two different pathways (Figure 2A). The most common pathway was coined non-amyloidogenic or anti-amyloidogenic due to its inability to produce amyloidogenic Aβ fragments [57,58]. In the non-amyloidogenic pathway, *APP* processing begins with the shedding of the ectodomain by a membrane-bound α-secretase of the ADAM family [59]. The main α-secretase for *APP* is ADAM10 [60], while studies suggest that TACE is the preferred α-secretase of *APLP2* [61]. The cleavage by the α-secretase is sequence-independent and occurs at a distance of 12-13 residues from the TM domain [62] to release the ectodomain (s*APP*α in the case of *APP*) and the membrane-anchored C83 fragment. In the amyloidogenic pathway, a β-secretase (BACE1) always starts the cleavage a few amino acids upstream of the α-cleavage site to release s*APP*β. BACE1 is an aspartyl protease that possesses some level of sequence specificity, although large discrepancies do exist between cleavages sites (for a review see [63]). This low level of sequence specificity results in two possible β-cleavage sites of *APP*. The one after Glu_682_ produces C89 but only the cleavage after Asp_672_ can generate the C99 that enables production of amyloidogenic Aβ peptides [64]. Mutations responsible for inherited AD are reported to promote the cleavage at Asp_672_ (i.e., the *APP* Swedish or *APP* KM670/671NL mutation) by providing a more favorable environment for β-cleavage at Asp_672_ [65,66]. The two other members of the *APP* family, *APLP1* and *APLP2*, are also processed by the β-secretase to produce C-terminal fragment (CTF) of 100 and 104 residues, respectively (Figure 2B). In both the amyloidogenic and non-amyloidogenic pathways, CTFs resulting from α- or β-cleavage are further processed by a γ-secretase complex. After initial ε-cleavage at position 48-49 (Aβ numbering in *APP*), which releases AICD and Aβ_48-49_ [67], the γ-secretase performs intramembrane successive cleavage of three (or four) residues at a time to produce C-terminal Aβ fragments ranging from 36 to 43 amino acids [68,69], of which Aβ_42_ and Aβ_43_ have the strongest oligomerization properties. This highlights the observation that the C-terminus of Aβ peptides is key for aggregation. Similarly, *APLP1* and *APLP2* are processed by γ-secretase complex but do not lead to the production of peptides with amyloidogenic properties [59,70], illustrating that Aβ-like oligomerization is a property specific to the *APP* sequence (i.e., transmembrane (TM) and juxtamembrane (JM) sequences) present in Aβ peptides. These sequences or motifs provide Aβ with fibrillogenic properties once it has been released upon γ-cleavage of C99. In addition to a specific C-terminal sequence, the pathologic properties of Aβ are also associated with the LVFF motif [71] (see below) present in the Aβ central region, which forms a β-hairpin and also plays a regulatory role in γ-secretase processing [72]. This motif is present only in *APP* and is not restricted to the C99 produced by β-secretase cleavage (Figure 2B), but is also present in C83. To date, the role of the N-terminus of Aβ sequence has been less investigated but it may impact the stabilization of Aβ aggregates [73]. In this respect, it is important to note that human and murine Aβ only differ by three amino acid residues in the N-terminal regions, and that murine Aβ is not amyloidogenic. Overall, growing evidence indicates that some peculiar motifs present in the human Aβ sequence (and not in related peptides generated by *APLP1/APLP2* processing) may play a role in amyloid pathology both (i) by regulating Aβ production and (ii) by promoting or preventing the assembly of Aβ building blocks into amyloid fibrils.

## 3. Transmembrane Interactions and Amyloidogenic Processing

The finding that *APP* forms dimers by homo- and hetero-interactions with other *APP* family members [41,74] was instrumental to the hypothesis that *APP* dimerization was a regulatory mechanism for its processing and function. Dimerization of *APP* was described early in the field and was first associated with its binding to heparin and collagen via the ectodomain [37]. Later, it was discovered that *APP* dimerization promoted the production of Aβ [74]. Homo-dimerization of *APP* was initially described between different segments of the ectodomain and for its role in cell-cell adhesion [41,75].

Transmembrane (TM) and juxtamembrane (JM) motifs involved in protein-protein interactions have attracted particular interest. Regarding the remarkable conservation of *APP* and *APLPs* TM sequences, these motifs may be involved in dimerization between all the members of the family. In addition to their involvement in dimerization, these motifs present in *APP* are found precisely in the Aβ sequence and in the vicinity of the *APP* cleavage sites releasing Aβ. Such motifs are GxxxG and GxxxG-like motifs, also known as Glycine zippers, and were described as key regulators of *APP* amyloidogenic processing [76,77]. GxxxG motifs were initially identified in the glycophorin A TM region [78,79,80,81]. The groove formed by an amino acid without a lateral chain (Gly) in an α-helix context allows close apposition of TM helices and packing into dimers by non-covalent interactions formed by amino acids surrounding the GxxxG sequence. Mutagenesis and structural analyses have revealed numerous examples in which the interaction between TM helices of single-pass membrane proteins is dependent on a GxxxG or (small)xxx(small) motif, where (small) designates amino acid residues with short side chains such as alanine. The interaction strength of motif-containing helices depends strongly on the sequence context and membrane properties. Several GxxxG-containing TM domains can interact via interfaces involving residues (hydrophobic, polar, aromatic) that are not organized in recognizable motifs. Importantly, in multipass membrane proteins, GxxxG motifs can be involved in protein folding, and not just oligomerization [82,83]. GxxxG and GxxxG-like motifs are enriched in protein TM regions, and strikingly, *APP* contains three in-register GxxxG motifs in its TM/JM region (Figure 3A), with an ADAD mutation (Flemish or A692G) adding a fourth motif in the JM sequence [84]. Such an occurrence of three GxxxG motifs is very rare (Figure 3B) and observed only in 17 human type I TM proteins (22 in *mus musculus*), and only one protein contains four GxxxG motifs in the region comprising the TM and six surrounding residues on each side. Intriguingly, the prion protein involved in prion disease by the misfolding of the endogenously expressed prion protein (PrPC) into an abnormal isoform (PrPSc) that has infectious properties, also contains a string of three in-register GxxxG motifs in its hydrophobic region [85]. However, the main differences are that PrPC is a membrane-anchored and not a TM protein, and that the hydrophobic regions—known as the hydrophobic tract—lie outside the α-helical C-terminal domain, and are therefore possibly in another conformation than the *APP* TM region. In the case of *APP*, GxxxG motifs are indeed involved in TM interactions once the bulky ectodomain has been removed by α- or β-secretase shedding (Figure 2A), and play key regulatory functions in the dimerization and processing of *APP* CTFs [75,76].

An important advance came from solid state Nuclear Magnetic Resonance (NMR) studies and other biophysical approaches applied to the JM-TM domains of *APP* using peptides resuspended in membrane-like bilayers, such as dimyristoyl-phosphocholine (DMPC): dimyristoylphosphoglycerol (DMPG) bilayers. Structural data indicated that the LVFF motif preceding the first GxxxG motif forms a β-sheet structure, that the GxxxG are likely helical, and that the helicity of the first GxxxG motif is enhanced by the A21G motif and by cholesterol [72,86]. Earlier NMR studies in detergent or detergent-lipid mixtures [87,88] showed a different picture for LVFF and GxxxG motifs that did not fit with processing. Careful titration of detergent by FTIR spectroscopy showed that detergent induced the helical structure of LVFF rather than the natural β-sheet. Interestingly, cholesterol appears to bind to the aromatic residues of the LVFF motif and influence the downstream structure [86]. Finally, introducing the Flemish mutation (A692G or A21G in Aβ numbering, see above) and adding cholesterol to the experimental context exerted additive effects on the helicity of the first GxxxG motif and on amyloidogenic processing [86]. Thus, *APP* processing at γ-sites might be controlled by specific orientations and interactions of its TM domain, with these processes (orientation and interaction) being controlled by the cellular context and, in particular, by the lipid composition of the membrane leaflets.

**Figure 3 biomedicines-10-02753-f003:**
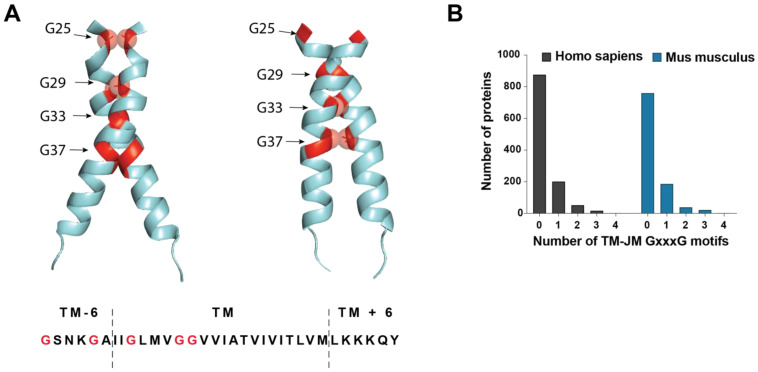
**The role of C99 dimerization in Amyloid β generation.** (**A**). Illustration of the role of GxxxG motifs in *APP* C99 dimerization. The figure illustrates the two dimerization interfaces described in [89]. The conformation shown on the left contains the ^25^GXXXG^29^ motif in the interface and was associated with production of SDS-resistant Aβ hexameric species. The conformation shown on the right interface contains the ^33^GXXXG^37^ motif in the interface and was linked to generation of AICD signaling properties and promotion of the Aβ_42_ processing line [89]. Glycines are shown in red, with the one in the interface shown as bubbles. (**B**). Occurrence of GxxxG motif in the TM-JM segments of type I TM proteins in the *homo sapiens* (black) and *mus musculus* (blue) proteome. Only one human protein contains 4 GXXXG in the TM-JM segments (not visible in the graph).

### 3.1. Dimeric Conformation of C99 Regulates γ-Secretase Processing

Interest for the dimerization of *APP* C99 occurred later with the observation that a large proportion of familial AD (FAD) mutations occurred in the TM-JM domain where TM interaction motifs are located. The observation that GxxxG motifs and dimerization of *APP* regulated Aβ production provided strong arguments for an influence of TM dimerization in its processing by γ-secretase. The role of *APP*/C99 dimerization in the regulation of γ-secretase processing was extensively investigated [77,90,91,92,93,94,95,96] but led to two contradicting observations. On one hand, dimerization was found to increase generation of pathogenic Aβ. For example, Scheuermann and colleagues designed stable *APP* homodimers by introducing the K699C mutation (K624C in *APP*_695_ numbering) at the extracellular JM-TM section ahead of γ-secretase cleavage sites and showed that dimerization dramatically increased Aβ production [74]. In line with these findings, compounds that reduced Aβ_42_ generation were shown to act by interfering with *APP* TM dimerization [97]. On the other hand, different lines of evidence suggested that forced *APP* TM dimerization was not compatible with γ-secretase processing. Using a system in which dimerization is imposed from the intracellular domain, Eggert and colleagues observed that controlled dimerization resulted in decreased Aβ generation [98]. More recently, however, the acquisition of the cryo-EM structure of γ-secretase in complex with C83 hinted that processing of C-terminal fragments (CTF) could indeed not occur in their dimeric form [99], in line with our own observations using a cell-free γ-secretase processing assay [89]. A first answer to these apparently contradicting observations was provided by our finding that GxxxG motifs were critical in regulating the orientations of TM dimerization and that a specific interface was required for amyloidogenic processing [76]. Using a system of fusion proteins that force dimerization of TM helices in all possible orientations, we demonstrated that precise dimeric orientations of C99 controlled γ-secretase processing by regulating the initial ε-cleavage and therefore influencing AICD-dependent signaling and generation of Aβ_42_ [89]. Notably, the dimeric orientation resulting in enhanced Aβ_42_ production and AICD-dependent signaling contained GxxxG motifs in the interface, but unfolding is mandatory for γ-processing since covalently bound *APP* CTFs are not further processed [89].

### 3.2. APP Dimeric Conformation Controls Its Intracellular Localization and Aβ Generation

In addition to its role in the regulation of processing, multiple lines of evidence indicate that dimerization of C99 is both a regulator and a consequence of its subcellular localization. The impact of *APP* dimerization on its subcellular localization was only recently investigated. A link between dimerization and subcellular localization was first suggested by Ben Khalifa and colleagues who observed that *APP* dimers were mostly present in the secretory pathway [75]. Using a model of forced dimerization of full-length *APP*, Eggert and collaborators analyzed in more detail the role of *APP* dimerization on subcellular localization and observed that *APP* dimerization resulted in increased localization in the endoplasmic reticulum (ER) and in endosomes, where the generation of pathogenic Aβ is believed to occur, by modulating its interaction with LRP1 and SorLA [100] (reviewed in [101]). The role of dimerization in intracellular trafficking was extended by Perrin and colleagues who observed that different dimeric orientations of C99 resulted in distinct subcellular localization in primary neurons and that the Aβ_42_ processing line was favored by a dimeric conformation that resulted in decreased cell surface and increased intracellular localization [89].

To better understand the impact of subcellular localization on amyloidogenic processing (see Figure 4), one must keep in mind that γ-secretase is composed of four subunits, namely, Nicastrin, Aph1, Pen-2, and Presenilin 1/2 (PS1/2, encoded by the *PSEN1* and *PSEN2* genes, respectively) [102]. The catalytic core of γ-secretase is composed of either PS1 or PS2, which harbor two aspartate catalytic residues (e.g., Asp287 and Asp387 for PS1) [103,104]. The fact that mutations in PSs are the most common cause of FAD highlights their critical role in the pathology (for a review see [105]). The link between subcellular localization of *APP* and PSs was recently investigated by Sannerud and colleagues [106]. Using a wide variety of approaches, they elegantly demonstrated that PS1 and PS2 have distinct subcellular localizations that impact C99 processing. More specifically, they observed that PS2 was addressed to late endosomes/lysosomes by its N-terminal dileucine sorting motif E_16_RTSLM_21_ via interaction with AP-1 while PS1 was more ubiquitously expressed, notably at the plasma membrane. As suggested previously [107], secreted Aβ peptides were increased in the absence of PS2 but the intracellular Aβ and Aβ_42_/Aβ_40_ ratio was higher in the absence of PS1. They further observed that cleavage of C99 in late endosomes was associated with the formation of aggregation-prone Aβ peptides that were less efficiency secreted. Remarkably, these results correlated with those obtained in two of our recent studies. In Perrin et al., we found that the dimeric orientation of C99 that favored the Aβ_42_ processing line and localized in intracellular compartments was more efficiently processed by PS2 [89]. In Vrancx et al., we reported that specific hexameric-like Aβ_42_ assemblies were mainly produced by PS2 in vesicular compartments [108]. The fact that PS2 favors the Aβ_42_ processing line in late endosomes must also be put in perspective with the observation that BACE1, the limiting enzyme of *APP* amyloidogenic processing, is predominantly found in endosomal compartments [109,110,111,112]. Together, these observations provide strong arguments for a crosstalk between C99 dimerization, subcellular localization, and generation of pathogenic amyloid β. It is also interesting to note that TM interaction motifs discussed above are not restricted to *APP*, but are also found in PS1 and PS2 [113], and in the Aph-1 subunit of the γ-secretase [114], and that the hydrophilic loop of PS1 together with the *APP* GxxxG TM motif regulates γ-secretase function in generating pathogenic Aβ peptides [115]. These observations indicate that the role of TM interaction in amyloid pathology may not be restricted to the substrate (*APP*), but may act all along the processing line controlling the orientation of *APP* dimers, the docking and fitting in the γ-secretase complex, and, eventually, the formation of pathogenic Aβ species.

### 3.3. The Role of Lipids in C99 Dimerization and Amyloidogenic Processing

Cell membranes are a buoyant environment and are largely heterogeneous in their composition. The link between AD and the lipid composition of membranes was previously suggested by Alois Alzheimer who observed “lipoid granules” and “adipose inclusions” as a third hallmark of Alzheimer’s disease [116,117]. A large body of evidence has since confirmed that lipid composition is indeed a critical factor influencing AD pathogenesis (reviewed in [118,119]). Cholesterol was early found to be a major risk factor of Late Onset Alzheimer’s Disease (LOAD) [120,121,122]. This argument is reinforced by the identification of the allele ε4 of ApoE, a major lipoprotein involved in cholesterol transport, as the most important risk factor of non-familial AD [123,124,125]. Early studies also indicated that low cholesterol promotes α-cleavage and non-amyloidogenic processing [126,127] whereas β-cleavage and amyloidogenic processing is favored by high cholesterol content [128] found in lipid rafts [129]. The role of cholesterol in C99 dimerization remains largely controversial. Different lines of evidence support the observation that C99 directly interacts with cholesterol [87,130,131]. Nuclear Magnetic Resonance (NMR) analyses revealed that interaction between cholesterol and C99 involved essentially the N-terminal loop of C99 and residues Glu693 and Asn698. Interestingly, glycines of the first GxxxG motif also seemed to directly bind cholesterol [131]. In line with the observation that GxxxG motifs interact with cholesterol, Song and colleagues suggested that cholesterol binding to C99 competed with homodimerization [132], and others showed that lowering cholesterol inhibited Aβ production by promoting *APP* dimerization [133]. These results are intriguing since a large body of evidence linked C99 dimerization to amyloid β production. Yet, other investigations tend to show that rather than preventing dimerization, cholesterol modulates the conformation of C99 dimers. Studies of Tang and colleagues indicated that the ^25^GxxxG^29^ motif was in fact stabilized by cholesterol [86], and others suggested that the membrane micro-environment and cholesterol, rather than competing with homodimerization of C99, had a role in the regulation of C99 dimeric conformation [134,135,136]. These studies differ notably by the composition of the membrane environment used, not only in the percentage of cholesterol, but also in other lipids that may impact C99 dimerization. One plausible scenario is that a moderately increased level of cholesterol promotes C99 dimerization in a particular conformation related to increased amyloidogenic processing. This conformation may target C99 dimers to lipid rafts where the very high cholesterol content induces destabilization of C99 dimers, thereby allowing processing by γ-secretase and generation of Aβ peptides (Figure 4).

An important question, directly related to lipid composition, concerns the thickness of the membrane. A thicker membrane will embed more residues in the juxtamembrane region and change tilt and ionization of the residues bordering the TM domain. The same sequence can also fold as an α-helix, random coil, or β-strands in different environments. The use of lipid formulations close to that of the cell membrane allows the conclusion that the GxxxG motifs are helical [86], yet in less physiological environments they form other structures [88], which is relevant for post-cleavage events and formation of oligomers. Moreover, prediction by replica-exchange molecular dynamics simulations of wild-type (WT) and mutant *APP* dimer conformations in which Gly residues from GxxxG motifs were changed to Leu illustrated large conformational differences in a membrane context. Dimerization of the WT (GxxxG) is due to two hydrogen bonds between two *APP* fragments, whereas dimerization of the mutant (LxxxL) is due to hydrophobic interactions. In the mutant, each *APP* fragment is more tilted, and the γ-cleavage site is shifted toward the center of the membrane. This position produces a mismatch between the active site of γ-secretase and the γ-cleavage site of *APP* that might impair Aβ production [137].

### 3.4. Impact of Familial AD Mutations on APP TM Dimerization: A Link with Aβ Production?

The fact that over 20 FAD mutations are located in *APP* TM domain illustrates the importance of this region for pathogenic Aβ generation. Yet, their influence on Aβ production through C99 dimerization remains quite elusive. Using synthetic peptides, Gorman and colleagues found that the V717G, V717F (Indiana), and T714I (Austrian) mutations all decreased TM dimerization and concluded that TM dimerization was negatively associated with pathogenic Aβ generation [138]. Similarly, others concluded that FAD mutations in the ^714^TVIV^717^ TM segments induced lower dimerization [139]. Conversely, So and collaborators analyzed dimerization of full-length *APP* FAD mutants with techniques including biomolecular fluorescence complementation (BiFC) and live cell crosslinking, and found that FAD mutations did not significantly affect the overall level of dimerization [96]. These observations may suggest that *APP* FAD mutations do not act by increasing C99 dimerization levels but may rather induce modulation of the C99 dimeric interface. Consistently, we observed that the *APP* Flemish mutation (A692G) promotes dimerization in a conformation that puts the GxxxG motif in the interface and increases Aβ_42_ production [86]. Strikingly, however, results obtained with split protein assays indicated that mutations of Gly residues present in *APP* GxxxG motifs did not affect the number of dimers formed, but rather generated specific Aβ assemblies [91]. This led to a new hypothesis: the TM interaction motifs (i.e., GxxxG or GxxxG-like) may indeed not only control dimerization and processing, but also the processes involved in Aβ assembly that lead to toxic aggregates. This hypothesis was reinforced by our most recent findings showing that forcing the ^25^GxxxG^29^ interface in dimeric Aβ_42_ and Aβ_43_ assemblies resulted in the generation of SDS-resistant hexameric Aβ species, while forcing dimerization in the ^33^GxxxG^37^ interface did not allow generation of such oligomers, even in the context of highly amyloidogenic Aβ_42_ and Aβ_43_ [89].

### 3.5. JM/TM Determinants Drive Aggregation Steps (Nucleation) Leading to Pathological Aβ Seeds

When the soluble Aβ (sAβ) peptides accumulate, they tend to spontaneously self-aggregate, eventually leading to the formation of fibrillar amyloid lesions. Over the past decades, a growing body of evidence indicated that Aβ intermediates, in particular soluble oligomers, are the primary cause of synaptic dysfunction and Aβ toxicity observed in AD. Indeed, while Aβ fibrils are large, insoluble materials aggregating into plaques, Aβ oligomers are soluble and may easily spread throughout the brain. Many oligomeric structures seem to play an important role in Aβ assembly and to have deleterious effects that may explain its related toxicity. For instance, dimers and trimers of Aβ have been consistently linked to long-term potentiation impairments [140,141,142]. Understanding the process of Aβ aggregation is crucial for identifying assembly steps that may be targeted by disease-modifying drugs.

Unlike the Tau protein, for instance, in which post-translational modifications (and especially phosphorylation) are closely associated with abnormal folding and aggregation, few post-translational modifications (PTMs) are described on Aβ and their possible role in Aβ aggregation remains uncertain. The Aβ peptide has been demonstrated to undergo several types of posttranslational modification, such as pyroglutamylation, N-terminal truncation, oxidation, glycosylation, nitration, isomerization, racemization, and phosphorylation [143,144]. However, whilst modified forms of Aβ are further investigated as potential markers of AD progression [145], such modifications are known to not be necessary for the induction of Aβ aggregation and subsequent deleterious effects.

One exception might be the phosphorylation of serine residue at position 8 and 26 (Aβ numbering), which has been reported to promote the formation of oligomeric Aβ assemblies or to stabilize them, respectively [146,147]. Thus, intrinsic determinants much more than PTMs are likely to be necessary for aggregation. This may be highly regulated by both (i) the nature of the C-terminal end and (ii) the presence of peculiar motifs that promote the folding and packaging of Aβ into oligomers and fibrils. The length of the Aβ peptide is correlated to its aggregation properties, with the longest forms (Aβ_42_ and Aβ_43_) being more hydrophobic and prone to assembling into oligomers and further elongating into fibrils. A faster rate of aggregation of Aβ_42_ in comparison to Aβ_40_ and a stable set of oligomers with a diameter distribution of ~7 to 9 nm was prevalently observed uniquely for Aβ_42_ even after fibril appearance [148]. Aβ assembly relies on a process called “nucleated polymerization”, involving three distinct phases: (i) a nucleation phase—or lag phase—during which several unfolded or partially folded monomers of Aβ come together to form an oligomeric nucleus; (ii) an elongation phase—or growth phase—in which the nuclei rapidly grow by further addition of monomers, giving rise to prefibrillar structures then ordered protofibrils; and (iii) a stationary phase—or equilibrium—where the concentration of monomers is low and constant, the assembly process reaches saturation, and protofibrils assemble into mature fibrils (Figure 5). The nucleation phase relies on Aβ monomers undergoing conformational changes and misfolding, which render it thermodynamically unfavorable, whereas the elongation phase is much more favorable and occurs rapidly. In primary nucleation, the initial formation of amyloidogenic nuclei occurs without contribution of pre-formed oligomers and constitutes the rate-limiting step. Amyloid fibril formation may also be seeded by the presence of preformed aggregates—oligomers, protofibrils, or fibrils [149]. As a consequence, a whole set of Aβ intermediate assemblies in Aβ fibril formation are present in the brain parenchyma. Some of them were shown to be readily formed in cells and having seeding properties when monomeric Aβ_42_ is available [108,150].

Peculiar motifs present in the Aβ sequence largely contribute to its complex aggregation properties. Strikingly, TM interaction motifs appear to play a key role not only in Aβ dimerization, but also in aggregation; in particular, the GxxxG motifs described above, which acquire aggregation properties after the transition from the α-helical structure found in the TM context to the β-sheet structure present in Aβ peptides. Glycines from GxxxG allow the turn in the β-helix and form a rigid and grooved structure in the β-sheet that allows stacking of amyloid fibrils [151]. It is tempting here to draw a parallel between GxxxG motifs involved in TM dimerization and their role in Aβ assembly by raising some key questions. Could Aβ stubs initially folded in an α-helical structure (corresponding to the *APP* TM structure) assemble into dimers that would initiate the nucleation process and promote the conversion to larger Aβ nuclei (Figure 5)? Some recent work supports this exciting hypothesis. First, specific *APP* TM orientations engaging a GxxxG interface lead to the production of specific (hexameric-like) Aβ assemblies. Remarkably, forcing a specific α-helical orientation of Aβ_42_ and Aβ_43_-like peptide dictates their ability to form oligomeric species [89]. Additionally, it has been recently reported that the modular Aβ C-terminal segment mediates rapid, non-nucleated formation of α-helical oligomers; as a result, high local concentration of tethered amyloidogenic segments within these α-oligomers facilitates transition to a β-oligomer population that ultimately generates mature amyloid [152]. Motifs located in the Aβ N-terminal segment are also involved in Aβ aggregation. This is particularly sound, since the N-terminal region of Aβ is present in the JM regions of its precursor (*APP*), which shows poor conservation across the *APP* family members. It therefore makes sense that intrinsic aggregation properties of Aβ directly relate to its N-terminal/central region. The LVFF motif (see above) plays a critical role in the Aβ assembly process. This LVFF sequence was initially proposed as an α-helix region that acts as a cholesterol sensor regulating *APP* processing [131,153]. Incorporation of cholesterol into model membranes enhances the structural changes induced by FAD mutations, suggesting a common link between familial mutations and the cellular environment. However, the LVFF motif was more recently shown to predominantly be a β-sheet in membrane bilayers that, when disrupted by changing lipid composition, increases Aβ production. In this β-sheet conformation, it can match hydrophobic motifs in Tau repeats and accelerate their aggregation via polymorphic states, appearing thus as a major interactor in both types of AD protein folding disorders [154].

**Figure 5 biomedicines-10-02753-f005:**
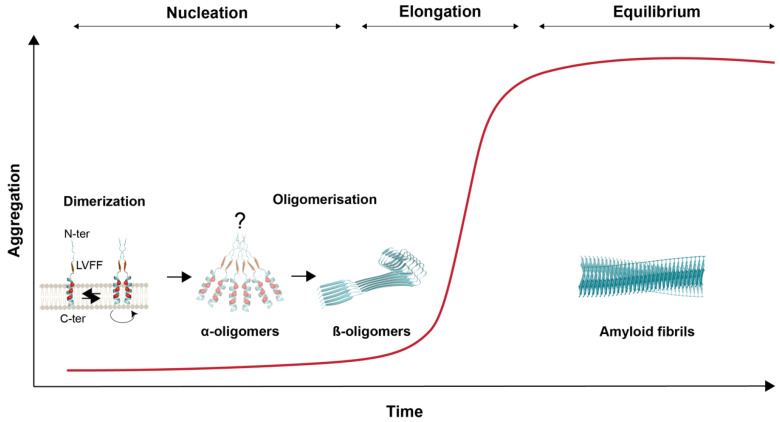
**Steps of amyloid β aggregation.** Illustration of putative nucleation steps of Aβ adapted from [152]. γ-secretase cleaves monomers of C99 (see above) but released Aβ peptides may rapidly reform dimers centered on the GxxxG motifs. In the nucleation phase, dimer formation is the initial step leading to the formation of higher-order species (hexamers are represented) whose nucleation is mediated by the Aβ C-terminus. These oligomers usually form β-sheets but may go through a transient step to form α-oligomers that rapidly switch to more stable β-oligomers initiated by the LVFF motif. Elongation occurs by assembly of β-sheet oligomers into fibrils.

## 4. Conclusions and Future Directions

Alzheimer’s disease is the first cause of dementia, and global ageing of the population is set to cause a dramatic rise in the number of cases, which is already very high. Despite recent controversy, the amyloid cascade hypothesis remains the most widely accepted in the scientific community given the robustness of its genetic and biochemical arguments. Yet, clinical trials targeting amyloid plaques have all failed to provide the expected benefits. These failures might be rooted in the fact that most trials have focused on targeting amyloid deposits rather than circulating Aβ oligomers, which are more believed to be the toxic species associated with the cognitive decline, although this hypothesis is also fiercely debated. Ultimately, the involvement of amyloid pathology in AD has been the subject of as much debate—if not more—as recent advances in knowledge.

Still, in recent years, important observations have advanced our understanding of the structural determinants that lead to the formation of toxic oligomers. The discovery that C99 dimerization and its dimeric orientations regulate its processing to generate pathogenic Aβ species was instrumental to understanding how amyloidogenic processing occurs. Recent advances highlight how different factors modulate the conformation of C99 dimers that, in turn, regulate amyloidogenic processing. Among them, membrane composition and, in particular, cholesterol content, seem to play a key role in the development of sporadic AD. The allele ε4 of ApoE, the main apolipoprotein involved in cholesterol homeostasis in the brain, is the principal risk factor for sporadic AD [123,155]. In addition, hypercholesterolemia was found to increase the production of Aβ oligomers in the brain of AD mouse models [156] and post-mortem analysis of the brain of AD patients demonstrated increased levels of cholesterol [157]. Despite this observation, the understanding of the role of membrane composition in the modulation of C99 dimeric conformations and generation of toxic amyloid species has been slowed and complicated by the heterogeneity of the methods used to recapitulate biological membranes.

Another critical question is whether the same motifs that regulate C99 processing and localization are also involved in the initial steps of Aβ oligomerization. Recent studies from two independent groups strongly suggest that early nucleation of Aβ could occur in a transient helical conformation, and that modulation of the dimeric orientation of helical Aβ-like peptides dictates the generation of Aβ oligomers [89,152]. This contrasts with the previous idea that Aβ oligomers are essentially formed of closely packed β-sheets. If this scenario holds true in future research, this would give additional weight to the role of GxxxG motifs and of membrane composition in the generation of toxic Aβ oligomers and the onset of Alzheimer’s disease. We propose that future studies should elaborate on these recent findings and assess how changes in membrane compositions of the ageing brain affect not only dimerization of C99, but also of Aβ peptides, and how this correlates with generation of toxic Aβ oligomers. Critically, it would be of major interest to assess how changes in lipid composition associated with ageing [158] affect the structural determinants of C99 processing and amyloid β oligomerization. We believe that understanding how the natural changes in membrane properties of ageing individuals associate with amyloidosis could indeed pave the way for a new array of therapeutic perspectives.

## Figures and Tables

**Figure 1 biomedicines-10-02753-f001:**
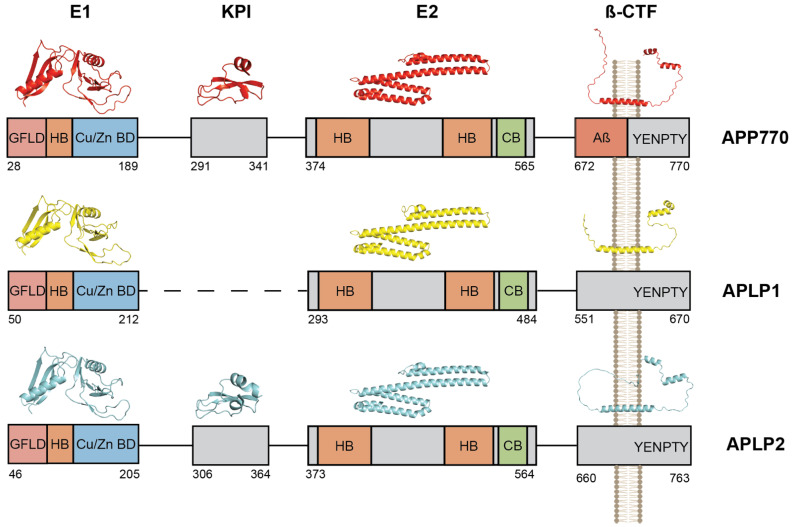
**Structure of the***APP***proteins family.** Structure of the *APP* and *APLPs* proteins predicted by AlphaFold 2.0 [34]. The structure of folded domains of each protein are shown as red, yellow, and blue for *APP*^770^, *APLP1*, and *APLP2*, respectively. Unstructured regions are not shown. The domains corresponding to each structure are detailed with amino acid numbering equivalent to the start and end of each subdomain. The YENPTY motif corresponds to the internalization motif conserved in all members of the *APP* family. HB: Heparin Binding (orange), CB: Collagen Binding (green), Cu/Zn BD (blue): Copper/zinc Binding Domain, GLFD: Growth Factor-Like Domain.

**Figure 2 biomedicines-10-02753-f002:**
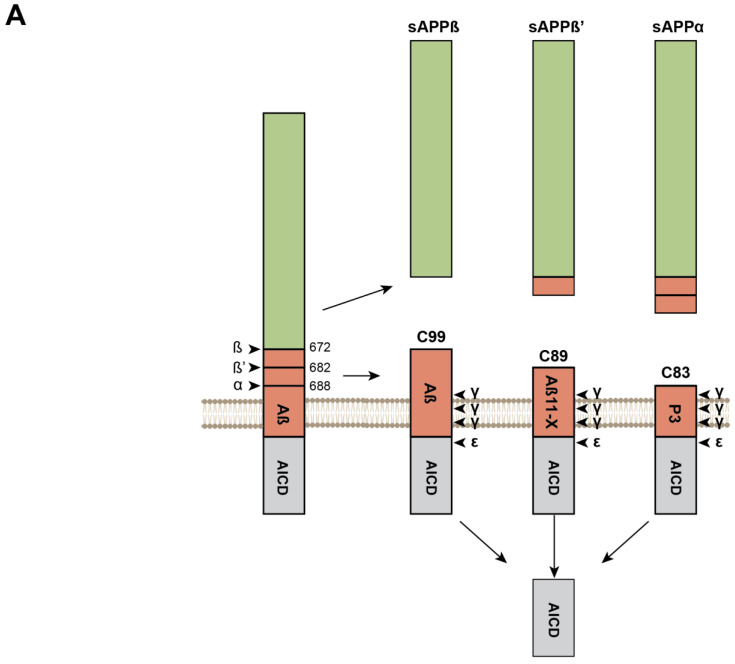
**Processing of the***APP***protein family.** (**A**). Illustration of *APP* processing. The sites of α, β, and β’ cleavage are shown with amino acids corresponding to cleavage position. Upon cleavage by α- or β-secretase, fragments called s*APP*α*,* s*APP*β and s*APP*β’ are released and secreted and C83, C89, or C99 remain embedded in the membrane, respectively. C99 contains the complete Aβ sequence, while C89 and C83 have truncated (11-X and 17-X, respectively) Aβ sequences. The Aβ 17-X is also called the P3 fragment. The first cleavage by γ-secretase occurs at the ε site to release AICD prior to intramembrane proteolytic cleavage of 3 or 4 amino acids to produce amyloid β (Aβ) peptides from 36 to 43 amino acids. (**B**). Sequence alignment of *APP* C99, C89, C83 and *APLP1*/2 β-CTF. The alignment illustrates sequence differences between the members of the *APP* family. The bottom panel highlights amino acids from most conserved (yellow) to less conserved (dark brown). The LVFF motif (red) is present only in *APP* while the C-terminal fragment is similar between *APP* and *APLP2*. The cleavage sites of γ-secretase are mostly conserved between all members of the *APP* family. The bottom chart illustrates the conservation of amino acid per position based on their properties with values ranging from 0 (no conservation) to 9 with * representing perfect conservation.

**Figure 4 biomedicines-10-02753-f004:**
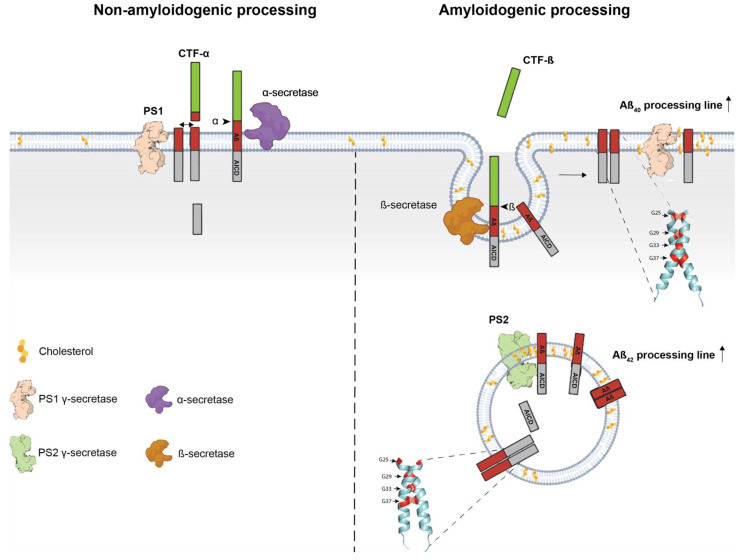
**The role of C99 dimerization in Aβ generation.** Illustration of the role of C99 dimerization, cholesterol concentration, and PSs localization on *APP*/C99 processing and Aβ generation. Initial cleavage by α- or β-secretase occurs at the plasma membrane or in endosomes, respectively. In addition, low cholesterol concentration favors α-cleavage while high cholesterol content is associated with β-cleavage. After initial cleavage, γ-secretase cleavage can occur either at the cell surface (preferably by PS1) or in endosomes (preferably by PS2). One hypothesis is that medium cholesterol concentration favors dimerization in one of the two dimeric orientations with either ^25^GxxxG^29^ or ^33^GxxxG^37^ motifs in the interface. Such dimeric concentration targets C99 either to endosomes (^33^GxxxG^37^ interface) or to the cell surface (^5^GxxxG^29^ interface). When concentration of cholesterol increases (such as in lipid rafts), dimerization is destabilized to allow processing by PS1- or PS2-dependent γ-secretase.

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
