# Peer review of "Structural Determinant of β-Amyloid Formation: From Transmembrane Protein Dimerization to β-Amyloid Aggregates"

_biomedicines, 2022, doi:10.3390/biomedicines10112753_

Round 1

Reviewer 1 Report

A broad summary of the subject is provided and different fields of the subject are well covered. Opinions are accurate and convincing. The review is very interesting, complete and very well structured.

A point could be corrected in my opinion, I think it will be useful for the reader to annotate what corresponds to the C83, C89, C99 directly in figure 2.

Page2, line 48 – LeV-ine

Page 5, line 180 – Asp672, 672 in subscript

Author Response

Reviewer 1

Thank you to the reviewer for reading the manuscript carefully and providing constructive criticism to improve it. For the sake of clarity, all changes mentioned are highlighted in yellow in the revised version.

Comments and Suggestions for Authors

A broad summary of the subject is provided and different fields of the subject are well covered. Opinions are accurate and convincing. The review is very interesting, complete and very well structured.

A point could be corrected in my opinion, I think it will be useful for the reader to annotate what corresponds to the C83, C89, C99 directly in figure 2.

Page2, line 48 – LeV-ine

Page 5, line 180 – Asp672, 672 in subscript

We have modified Figure 2 accordingly and corrected the mistakes in the text.

Reviewer 2 Report

In this review, the process of amyloid β peptide formation is discussed in detail. In general terms, the objective set by the authors is satisfactorily fulfilled.

Nevertheless, the biggest problem is the introduction, where the authors decided to talk about "proteinopathy" in general terms and generalize information that may be true for Aβ, but that definitely is not true for all diseases (or has not been confirmed).

- If the term proteinopathy is the same as “protein conformation disorders” or “protein folding diseases”, why do we need it?.

- Does Proteinopathy include a diseases where the “protein” related is not “mis-folded”? (It is possible to have a no-working protein that nevertheless it has the “correct” folding).

- Proteinopathy (= protein conformation disorders), is not the same as “proteins that forms amyloids”. There are several protein conformation disorders where there is not formation of amyloid fibrils.

Line 32. Each “Proteinopathy” is different; it is related to a different protein, in different environments, etc. Seems really unlikely that the concept of, for instance, “Aβ production essentially by regulating membrane dimerization” can be “transposed to the study of other proteinopathies”.

Line 44: Are you sure that “for all, abnormal folding and aggregation is considered to be a primary event in disease onset and progression”? References?

Line 55: Not all the confomationl diseases form β-sheet structures; All the “amyloids” diseases does it.

Lines 58: Amyloiis are defined as unbranched fibrils. To which one are you referring in the sentence “forming generally unbranched fibrils”?

Line61: False: “formation of amyloid fibrils is always preceded by the generation of soluble oligo-

meric intermediates that are believed to be key drivers of cellular toxicity”. Not always.

Other:

  • Line 124: E1 is composed of GFLD and CuBD segments but in Figure 1 only CuBD is represented.

  • Line 163: Aβ aggregates is joined (Aβaggregates).

  • Line 282: the word Amyloid is misspelled (Amlyloid).

  • Line 282: The dimerization does not occurred for C83 / C89?.

  • Line 290: Separate the word GXXXG in (GXXXGin).

  • Line 378: PS refers tp PSEN?

Figure 4. The separations in the figure and its captions are not clear. Maybe use different panels. One for the non-amyloidogenic a). other b) for the amyloidogenic. And c) fo the ...

  • Line 449 and 318: Aβ42 of Aβ42 , etc.

  • Line 525: (sgement).

  • Line 558: missing “.”

  • Line 575: in.

Author Response

Reviewer 2

Thank you to the reviewer for reading the manuscript carefully and providing constructive criticism to improve it. For the sake of clarity, all changes mentioned are highlighted in yellow in the revised version.

Comments and Suggestions for Authors

In this review, the process of amyloid β peptide formation is discussed in detail. In general terms, the objective set by the authors is satisfactorily fulfilled.

Thanks for the positive comment.

Nevertheless, the biggest problem is the introduction, where the authors decided to talk about "proteinopathy" in general terms and generalize information that may be true for Aβ, but that definitely is not true for all diseases (or has not been confirmed).

We agree with the reviewer on this critical point and we have modified our manuscript to address this concern, as detailed below. In the first version, the term proteinopathy has been too much misused as a shortcut to illustrate what was indeed protein folding diseases or protein folding disorders. It has been corrected, accordingly

- If the term proteinopathy is the same as “protein conformation disorders” or “protein folding diseases”, why do we need it?.

For the sake of clarity, we have removed the term “proteinopathy” and we have used the term “protein folding diseases” (see above).

- Does Proteinopathy include a diseases where the “protein” related is not “mis-folded”? (It is possible to have a no-working protein that nevertheless it has the “correct” folding).

Under the term “protein folding diseases” that we now use throughout the manuscript, we only include diseases caused by proteins acquiring pathogenic conformations.

- Proteinopathy (= protein conformation disorders), is not the same as “proteins that forms amyloids”. There are several protein conformation disorders where there is not formation of amyloid fibrils.

We agree with the reviewer on the fact that not all protein folding diseases form amyloids, but the largest group of protein folding diseases exhibit the formation of amyloids (Chiti et al, 2006).  For clarity, we have added the following sentences in the introduction: “Indeed, a great number of protein folding diseases are associated with the formation of highly organized fibrillar aggregates often described as amyloids [4]. This group of protein folding diseases is also known as amyloidoses. However, it is worth noting that some protein folding diseases are characterized by protein aggregates that are not amyloid in nature.”

Line 32. Each “Proteinopathy” is different; it is related to a different protein, in different environments, etc. Seems really unlikely that the concept of, for instance, “Aβ production essentially by regulating membrane dimerization” can be “transposed to the study of other proteinopathies”.

Again here this was due to the unprecise use of the term of proteinopathy, that has been fixed throughout the manuscript. Regrading the introduction, the sentence in question has been modified to more clearly convey the idea that molecular determinants and the cellular context both control the pathophysiological conformational transitions of a protein, APP being taken as an example. “This concept can be transposed to the study of other protein folding disorders, to better decipher the patterns switching protein conformation from physiological to pathological and improve our understanding of these mechanisms that devastate neuronal functions.”

Line 44: Are you sure that “for all, abnormal folding and aggregation is considered to be a primary event in disease onset and progression”? References?

It has been considered that abnormal protein folding and aggregation play a causative role in those diseases (as supported by references 1-3). We have included the appropriate references and clarified the sentence: “For the vast majority all of them, abnormal protein folding and aggregation are considered to play a causative role [1-3].”, to temper the concept of wholeness which can always be debated

Line 55: Not all the confomationl diseases form β-sheet structures; All the “amyloids” diseases does it.

This is true. The sentence has been modified: “In particular, the disease-causing conformational changes in amyloidoses lead to an enrichment in β-sheet structures, which are associated with an increased propensity to form amyloids.”

Lines 58: Amyloiis are defined as unbranched fibrils. To which one are you referring in the sentence “forming generally unbranched fibrils”?

We have modified the sentence to make it clearer: “These insoluble protein aggregates, regardless of the source of the amyloid protein/peptide, share a common fibrillar configuration with a unique structural core.”

Line61: False: “formation of amyloid fibrils is always preceded by the generation of soluble oligomeric intermediates that are believed to be key drivers of cellular toxicity”. Not always.

The term “usually” has been added to address this concern.

Other:

Line 124: E1 is composed of GFLD and CuBD segments but in Figure 1 only CuBD is represented.

This is corrected.

Line 163: Aβ aggregates is joined (Aβaggregates).

This is corrected.

Line 282: the word Amyloid is misspelled (Amlyloid).

This is corrected.

Line 282: The dimerization does not occurred for C83 / C89?.

Studies on dimerization of C-terminal fragment (CTFs) of APP has focused on C99 due to its role as the precursor of Aß which is relevant for the disease. It is in fact likely that dimerization of C83 and C89 occurs and regulates its processing but this has never been adequately studied. Indeed, our previous studies (Decock et al., FEBS Open Bio 2015) carried out with fusion proteins (split luciferase) showed that dimerization of C83 occurs. Still, the outcome in terms of processing had to be addressed by AICD release (including reporter gene assays), since C83 does not produce an amyloidogenic by-product (p3). For these reasons, we did not discuss it in this review.

Line 290: Separate the word GXXXG in (GXXXGin).

This is corrected.

Line 378: PS refers tp PSEN?

We homogenized to talk about PS when we refer to proteins and PSEN when referring to the genes, a commonly accepted use for both terms.

Figure 4. The separations in the figure and its captions are not clear. Maybe use different panels. One for the non-amyloidogenic a). other b) for the amyloidogenic. And c) fo the ...

We modified the figure for more clarity. We believe it is best to represent it as one single figure but we separated more clearly amyloidogenic and non-amyloidogenic processing.

Line 449 and 318: Aβ42 of Aβ42 , etc.

This is corrected.

Line 525: (sgement).

This is corrected.

Line 558: missing “.”

This is corrected.

Line 575: in.

This is corrected.

Reviewer 3 Report

The paper is excellently organised and useful to the area of research.

The contribution is excellent and adds new knowledge.

The amyloid aggregation information is complete and explained more with Figures. 

Author Response

Reviewer 3

Thank you to the reviewer for the critical reading of the manuscript concluded by very positive comments.

The paper is excellently organised and useful to the area of research.

The contribution is excellent and adds new knowledge.

The amyloid aggregation information is complete and explained more with Figures.
